# Public Perspectives on Exposure Notification Apps: A Patient and Citizen Co-Designed Study

**DOI:** 10.3390/jpm12050729

**Published:** 2022-04-30

**Authors:** Esli Osmanlliu, Jesseca Paquette, Maria Alejandra Rodriguez Duarte, Sylvain Bédard, Nathalie de Marcellis-Warin, Majlinda Zhegu, Catherine Régis, Marie-Eve Bouthillier, Annie-Danielle Grenier, Paul Lewis, Marie-Pascale Pomey

**Affiliations:** 1Research Institute of the McGill University Health Centre, Montreal, QC H4A 3J1, Canada; 2Division of Pediatric Emergency Medicine, Department of Pediatrics, Montreal Children’s Hospital, McGill University Health Centre, Montreal, QC H4A 3J1, Canada; 3Research Centre of the University of Montréal Hospital Centre, Montreal, QC H2X 0A9, Canada; jesseca.paquette@umontreal.ca (J.P.); maria.alejandra.rodriguez.duarte@umontreal.ca (M.A.R.D.); marie-pascale.pomey@umontreal.ca (M.-P.P.); 4Centre of Excellence for Partnership with Patients and the Public (CEPPP), Montreal, QC H2X 0A9, Canada; sylvain.bedard@ceppp.ca; 5Department of Mathematics and Industrial Engineering, Polytechnique Montreal, Montreal, QC H3C 3A7, Canada; nathalie.demarcellis-warin@polymtl.ca; 6Center for Interuniversity Research and Analysis on Organizations (CIRANO), Montreal, QC H3A 2M8, Canada; 7Department of Management, University of Québec in Montréal, Montreal, QC H2X 1L7, Canada; zhegu.majlinda@uqam.ca; 8Law Faculty, University of Montreal, Montreal, QC H3T 1J7, Canada; catherine.regis@umontreal.ca; 9Office of Clinical Ethics, Faculty of Medicine, University of Montreal, Montreal, QC H3C 3J7, Canada; mebouthillier.csssl@ssss.gouv.qc.ca; 10DIGICIT Advisory Committee, Research Centre of the University of Montréal Hospital Centre, Montreal, QC H2X 0A9, Canada; adgrenier@icloud.com (A.-D.G.); paul.lewis@umontreal.ca (P.L.); 11Department of Health Policy, Management and Evaluation, School of Public Health, University of Montreal, Montreal, QC H3N 1X9, Canada

**Keywords:** survey, COVID-19, app, exposure notification, citizen, patient, perceptions

## Abstract

Canada deployed a digital exposure notification app (COVID Alert) as a strategy to support manual contact tracing. Our aims are to (1) assess the use, knowledge, and concerns of the COVID Alert app, (2) identify predictors of app downloads, and (3) develop strategies to promote social acceptability. A 36-item questionnaire was co-designed by 12 citizens and patients partnered with 16 academic researchers and was distributed in the province of Québec, Canada, from May 27 to 28 June 2021. Of 959 respondents, 43% had downloaded the app. Messaging from government sources constituted the largest influence on app download. Infrequent social contacts and perceived app inefficacy were the main reasons not to download the app. Cybersecurity, data confidentiality, loss of privacy, and geolocation were the most frequent concerns. Nearly half of the respondents inaccurately believed that the app used geolocation. Most respondents supported citizen involvement in app development. The identified predictors for app uptake included nine characteristics. In conclusion, this project highlights four key themes on how to promote the social acceptability of such tools: (1) improved communication and explanation of key app characteristics, (2) design features that incentivize adoption, (3) inclusive socio-technical features, and (4) upstream public partnership in development and deployment.

## 1. Introduction

### 1.1. Background

Manual contact tracing and exposure notification is a key containment strategy in response to emerging outbreaks. However, it is resource-intensive and subject to imperfect recall of recent contacts. Therefore, the high transmissibility of SARS-CoV-2, including in its pre-symptomatic stages, has posed significant challenges to manual contact tracing worldwide [1]. Exposure notification apps can support manual contact tracing by digitally collecting information within the network of users. Such tools may therefore enable targeted confinement strategies and ultimately help reduce pathogen transmission [1]. Although the term “contact tracing app” is typically used in academic texts and the general press, the primary purpose of such tools is to rapidly identify and notify individuals who have had a high-risk exposure [1]. Terminology matters, as it may erode public trust in tools that are perceived to enable state surveillance of individual mobility (e.g., tracing, tracking). In this context, the contact tracing technology codeveloped by Apple and Google refers to an “Exposure Notification System” [2]. Similarly, the Government of Canada encourages the public to use COVID Alert, an “exposure notification app” [3]. We therefore use this term throughout the text.

Most studies have suggested a high public willingness to adopt exposure notification apps during the COVID-19 pandemic [4,5,6,7,8,9,10,11]. However, the available data suggest relatively low rates of use in practice, particularly in countries where the app is voluntary. For instance, downloads as of April 2021 ranged from 13% in India to 45% in Finland [12]. In contrast, more than 90% of the population had downloaded the mandatory TraceTogether app in Singapore [13]. In Canada, adoption rates averaged 23% [14]. In Québec, between March and July 2021, the proportion of app users was even lower (8%) [15]. Initial mathematical models suggested that sufficient app effectiveness required adoption by at least 60% of the population [16]. Subsequent models demonstrated that any level of uptake may help lower viral transmission [17]. Although data on real-life effectiveness remains scarce, a British study found a non-negligible epidemiological impact in England and Wales, where 28% of the population used the app regularly [18]. A modeling study estimated a much lower impact in Québec, with an estimated 300–600 infections and 3–6 deaths averted by the app [15]. The province has been severely affected by the pandemic, particularly during the first wave of infections, when it experienced the most cases per capita and highest case fatality rate in Canada [19]. A widely adopted exposure notification tool could have helped slow the massive surge of cases observed in multiple waves since the onset of the pandemic, thus alleviating the strain on hospitals and reducing mortality.

COVID Alert’s failure to launch, as noted for other exposure notification apps as well, may seem surprising given the initial interest in this technology and the substantial threats to population health and civil liberties that the COVID-19 pandemic has generated globally. Several aspects can affect the adoption of this digital innovation, including installation regimes, data management strategies, underlying technologies, digital literacy, and user perceptions. For instance, fears over data privacy, breaches in confidentiality, and mass surveillance are among the main concerns that users raised in previous surveys [4,5,6,7,10,11]. Furthermore, baseline Internet usage [20], demographic characteristics, and user perceptions of the app also affect adoption rates [11,21]. This situation highlights the complex interplay between environmental factors, personal characteristics, and user perceptions that ultimately affect the adoption of such tools, even in a pandemic context marked by a sense of urgency and the public’s desire to reach a post-pandemic reality.

With the epidemiology of SARS-CoV-2 evolving and the risk of future pandemics still present [22,23], exposure notification apps may play a role in fighting future outbreaks [24]. Much remains to be learned about this technology, as digital health innovations continue to emerge and reshape the public health landscape. The Canadian federal government formed a trans-disciplinary advisory council to evaluate the COVID Alert app and determine whether it can evolve to meet the needs of both the population and public health authorities. One of the council’s recommendations was to study reasons for non-adoption [25]. The experience gained through the introduction of this tool during the COVID-19 pandemic can also inform the responsible development and deployment of future digital health interventions.

Few studies have reported public engagement in the design or deployment of this digital innovation. Most cases of engagement were limited to cross-sectional consultations [26]. To our knowledge, no previous study has assessed public perspectives on exposure notification apps by engaging patients and members of the public upstream in the research process. This partnership between potential technology users and a team composed of 16 academic researchers and collaborators allowed the inclusion of key study themes that may have otherwise been overlooked [27].

### 1.2. Objectives of This Study

The study object was the COVID Alert app, from the perspective of Québec residents. Given the existing knowledge gap on the adoption of digital exposure notification apps, our study addressed three objectives. First, it assessed the use of the COVID Alert app (Ottawa, ON, Canada) by evaluating app adoption, user knowledge concerns, and the role of upstream citizen engagement in development. Second, it identified predictors of app downloads according to individual characteristics and perceptions regarding the innovation. Lastly, based on these results and the existing literature, it developed strategies that may promote the social acceptability and continuous use of exposure notification apps.

## 2. Materials and Methods

This study is reported in accordance with the SUrvey Reporting GuidelinE [28]. It was conducted in three steps: (1) the co-construction of a questionnaire, (2) the distribution of the questionnaire in the province of Québec, and (3) the statistical analysis of the data and its interpretation in partnership with patient and citizen collaborators.

### 2.1. The COVID Alert App

The COVID Alert exposure notification app was introduced in nine Canadian provinces and one territory in July 2020 [3]. It is a voluntary, decentralized app (i.e., data are stored in user’s smartphone as opposed to a centralized server) that uses Bluetooth technology to exchange random, anonymous codes between smartphones [3,29]. To download the app, the user is asked to choose one of the country’s two official languages (English or French) and mandatorily read information on the app’s contribution to the fight against the COVID-19 pandemic, the technology used, and data confidentiality. Once the user agrees to app activation, a green logo appears along with information regarding any recent exposure. Canadian provincial and territorial governments bear most of the responsibility for delivering health and social services to their populations [30]. Therefore, the local operationalization of the COVID Alert app varied across the country. In the province of Québec, app users who tested positive for SARS-CoV-2 could obtain a confidential one-time key by calling a designated provincial health line. They were then asked to enter it in the app within 24 h, which confidentially alerted their high-risk contacts from the preceding 14 days. A significant exposure between two app users consisted of a close contact (within 2 m) lasting at least 15 min.

### 2.2. Co-Construction of the Questionnaire

An advisory committee with citizens and patients was created by the research team, to co-construct a questionnaire to measure app adoption and user concerns regarding the COVID Alert app. Two virtual co-construction sessions were held. The first focused on how to formulate the questions (*n* = 84), and the second was to prioritize, rephrase and, in some cases, eliminate them. Four drop-in sessions were held in which the members of the advisory committee and the research team helped finalize the list of questions. French and English versions were created. The questionnaire was professionally adapted to promote readability at a high school level.

For this survey, we relied on face validation from the advisory committee and the research team. The final version of the questionnaire (Appendix A) contained 36 questions, in addition to 8 sociodemographic questions. Three concepts were incorporated into the questionnaire: (1) the use of a smartphone and the COVID Alert App (access to a smartphone, experience with the COVID Alert app, and influences on the decision of whether or not to download it), (2) perspectives on the COVID-19 pandemic and exposure notification apps (motivations and barriers to the decision to download it or not, the perceived personal health risk with respect to SARS-CoV-2, and the role of citizen involvement in the development of the app), and (3) respondents’ knowledge and concerns about the COVID Alert app. Fifteen questions were yes/no questions, fourteen were answered on either a four- or a five-point Likert scale, four were multiple-choice questions, and three were open-ended questions.

In the absence of an objective measure of app download, we selected self-reported download as a proxy for app adoption. Indeed, once downloaded, COVID Alert operates in the background without requiring the user’s active participation. We also asked those who downloaded the app whether they later uninstalled it, to assess the level of retention. Moreover, since infected app users in Québec had to voluntarily input a one-time key into the app to confidentially notify their recent high-risk contacts, we asked respondents whether they had done this, or would have, in the event of a positive test.

Candidate predictors of app download were identified through a thorough synthesis of the literature on prior surveys [26] and group discussions within the research team. The predictors included sociodemographic characteristics, baseline health status, as well as knowledge and perceptions of digital exposure notification.

The pretest of the questionnaire was held from 17 to 21 May 2021 among the members of the advisory committee and research team. Modifications mostly consisted of reformulating items to improve understanding.

### 2.3. Survey Administration

An independent research firm, SOM, handled participant recruitment and data collection, which was conducted from 27 May to 28 June 2021. Figure 1 describes the epidemiological context in the province of Québec during that period. The eligibility criteria included: (1) age ≥ 18 years old, (2) residence in the province of Québec, and (3) fluency in French or English. To include perspectives across different levels of digital literacy, the survey was administered online or by telephone to individuals who were part of SOM’s panel of respondents. No incentives were offered. The sampling plan aimed for 850 online and 150 phone surveys. The study sample was drawn using an algorithm that sought to promote representativeness of the province according to region, age, gender, primary language, education, and household size. For the web data collection, respondents were recruited randomly through the firm’s probabilistic panel. The sample for phone surveys was generated by targeting panel members who do not use the Internet. The estimated sample size ensured estimates with a margin of error of +/− 3.8% while ensuring sufficient representation from these subgroups.

The study protocol was approved by the Research Ethics Board of the University of Montréal Hospital Centre (CHUM; approval number: 2021-9202, 20,276, 17 November 2020). Consent was explained in the invitation message and was obtained when participants chose to answer the survey (Appendix A).

### 2.4. Statistical Correction and Analysis

Only fully completed questionnaires were analyzed. Survey weights were applied to make the sample of survey respondents more representative of the study population (Québec adult population), taking into account the following variables: (1) age distribution (18–34 years, 35–44 years, 45–54 years, 55–64 years, and 65 years or older), region (Montréal CMA, Québec City CMA, and elsewhere in Québec) and gender, (2) primary language (French, English, and other), (3) highest diploma or certificate held (with or without a university diploma/other), and (4) proportion of Internet users by region. Data from the Institut de la statistique du Québec was used for the age and gender distributions, and from the 2016 census for the other variables. A multivariate weighting procedure, consisting of 10 iterations of iterative proportional fitting, was carried out to reflect all of these distributions. Data were first processed by SOM using the MACTAB specialized software package (Québec, QC, Canada) and then analyzed by one of the authors (M.A.R.D.) using RStudio Version 1.3.1093 (Boston, MA, USA; details on modelling packages and coding can be found in Appendix A).

Weighted descriptive statistics are presented for the overall study population. Descriptive and regression analyses that presuppose smartphone ownership (e.g., decision to download the COVID Alert app or not) pertain to the subpopulation of respondents who owned a smartphone. For the three open-ended questions, free-text answers were coded thematically and reported accordingly.

A multivariable logistic regression model was used to identify the predictors of app download. The specifications of the model and the coefficients can be found in Appendix A. A similar approach was taken to identify factors associated with user perceptions and knowledge of the app. Additional models were estimated to evaluate the internal consistency of items related to user concerns, as a measure of internal validity. Missing data were excluded from the regression analyses.

## 3. Results

### 3.1. Respondent Characteristics

A total of 2249 invitations were sent by email to eligible respondents. The access rate was 41.4%, and the response rate was 38.2% (more details are provided in Appendix A). For the telephone interview invitations, the initial sample was 844. The response rate was 28.5%, the non-response rate was 30.8%, and the refusal rate was 40.7%. In total, 959 adults agreed to participate: 859 via the web survey and 100 via the telephone survey. A summary of the demographic characteristics of the respondents before weighting is presented in Table 1.

### 3.2. App Adoption and User Characteristics

Among the participants who owned a smartphone (*n* = 791), 43% had downloaded the COVID Alert app. Among 15% of those who had initially downloaded the app, it was later de-activated or uninstalled. Moreover, 38% of users who had received a positive COVID test did not enter the information in the app (*n* = 4). In contrast, among the users who had not received a positive COVID test, 93% reported that they would share it confidentially through the app. Table 2 presents the descriptive statistics on the respondents’ experience with and perspectives on the COVID Alert app and the COVID-19 pandemic.

Most respondents who downloaded the app frequently used other apps that collect personal data (83%). Concerning app literacy, 43% reported a low level of knowledge of the COVID Alert app. The most significant sources of influence in the decision to download the COVID Alert app were the Government of Québec (32%) and the Government of Canada (31%). Among those who did not download the app (53%), the most frequent reasons that would have motivated them to download it were having more evidence on app effectiveness (23%) and having more social contacts (20%).

### 3.3. User Concerns

Overall, 38% of users had concerns about the COVID Alert app. The main areas identified were cybersecurity (18%), fears over loss of privacy (18%), confidentiality (17%), and geolocation (14%). Other concerns included a perceived lack of effectiveness (6%), the use of data for other purposes (e.g., commercial; 5%), a lack of overall confidence in the app (2%), and a lack of information received about it (1%). Regarding privacy concerns, 46% believed that the COVID Alert app locates users by GPS (global positioning system) and 29% feared that it can identify users.

### 3.4. Predictors of App Download

In the study population, women were more likely to download the COVID Alert app (OR 2.26, 95% CI 1.36–3.76). Moreover, perceived app literacy is positively associated with app download. In particular, those who self-rated their knowledge of the app as average (OR 3.49, 95% CI 1.98–6.14) or high (OR 5.27, 95% CI 2.35–11.82) were more likely to download it. Trust in the fact that the app only collects information that the user consented to provide (OR 2.28, 95% CI 1.06–4.89) was positively associated with app download.

On the other hand, francophone respondents (OR 0.41, 95% CI 0.20–0.87), people not employed (OR 0.11, 95% CI 0.02–0.43), and those who reported “other” as their occupation (OR 0.20, 95% CI 0.04–0.92) were less likely to download the app. People who did not think that an app such as COVID Alert should be used for other health emergencies (OR 0.24, 95% CI 0.11–0.50) were less likely to download it. The perceived stress that the app could cause (OR 0.25, 95% CI 0.14–0.45) and the perception that it may clog up the health care system (OR 0.28, 95% CI 0.15–0.60) reduced the likelihood of downloading the app. Finally, people who rarely used other common apps (OR 0.33, 95% CI 0.14–0.76) were less likely to download the COVID Alert app. Figure 2 shows the adjusted odds ratios and their respective confidence intervals for the predictors of application downloading.

### 3.5. Characteristics Associated with User Perceptions and Knowledge of the App

Concerns regarding user identification through the COVID Alert app varied across age groups. Respondents aged 25–34 years (OR 4.55, 95% CI 1.08–20.28) were most likely to hold this belief than the group of 19–24 years old. Those who believed that the app could be used to monitor the population (OR 3.27, 95% CI 1.84–5.82) were also more likely to indicate that the COVID Alert app can identify users. In contrast, those who did not think that the app uses GPS technology (OR 0.26, 95% CI 0.14–0.48) were less likely to indicate that it can identify users.

As for opinions regarding privacy and data security, age was negatively associated with the idea that the app protects personal data. Adults 25–34 years old and those 34–44 years old were less likely to think that the app protects personal data, compared with adults aged 18–24 years old (OR 0.15, 95% CI 0.02–0.80, OR 0.11, 95% CI 0.02–0.53, respectively). However, those who have a high level of knowledge of the app (OR 3.07, 95% CI 1.37–6.90) were more likely to have this perception.

In exploratory regression analyses, the question related to general app concerns showed good internal consistency with other items assessing individual concerns, such as privacy or population surveillance (Appendix A).

### 3.6. User Recommendations for App Improvement and the Role of Citizen Engagement

Respondents identified design characteristics as the main area where the app could be improved (45%). In open-ended questions, some recommended adding local statistics (the number of people encountered, when they were exposed, etc.), incorporating information on how to obtain and enter the one-time key in the app, having the positive test automatically recorded in the app, or integrating the tool into a vaccination passport. Moreover, 66% of respondents indicated that citizens should be involved in the development of this type of application, and 34% were interested in participating personally.

## 4. Discussion

The first objective was to measure app adoption and user concerns over the COVID Alert app. The results suggest a high adoption rate in the study population, as over 40% of respondents had reportedly downloaded the app. However, more than 1 in 7 respondents later de-activated or uninstalled it. In comparison, 4 out of 5 respondents frequently used other personal data-collecting apps, such as social media and search engines. These results support the privacy paradox, as most people use these other apps, but less than half of respondents had downloaded the COVID Alert application despite its privacy-protecting features. The privacy calculus [32], where users do not perceive enough benefits in a tool, may have contributed to this gap. Moreover, among the few of respondents who received a positive COVID test, over 1 in 3 did not enter a one-time token in the app to confidentially alert their high-risk contacts.

Regarding the decision on whether to download the app or not, users identified government sources as a main influence. Following an initial media campaign in the first weeks of deployment, government references to COVID Alert dwindled, similar to daily download counts nationally [14]. In January 2022, at the peak of the fifth pandemic wave, a popular Québec newspaper reminded its readership that the app still existed [33]. Among non-users, having more contact with other people and having more evidence of the app’s effectiveness would have motivated them to download it. Doubts regarding the app’s effectiveness were also found to be one of the main reasons for non-adoption in prior studies [21,34]. Data protection and privacy were among the most frequent concerns reported, which is congruent with previous surveys [4,5,6,7,10,11]. As shown in other settings [35,36], limited trust in the government may have heightened privacy concerns, thereby affecting app download. Moreover, half of the respondents perceived themselves to be at a low risk of exposure of COVID-19. As suggested in the Health Belief Model [37], relatively limited app download is unsurprising given this level of low threat perception.

Finally, suggestions on how to improve the COVID Alert app included adding information in the app, such as statistics on the pandemic, the current sanitary measures to follow, the person’s vaccination status, and the results of their COVID-19 tests. This kind of knowing-by-using type of knowledge highlights the importance of involving citizens in the development of these types of apps in the future, in order to better identify important features to be included that could translate into higher uptake by users.

Our second objective was to identify predictors of app download related to individual characteristics and perceptions of this innovation. We found several predictors of app download related to user characteristics and perceptions. Notably, a high self-reported level of app literacy was positively associated with download. A previous survey identified this as the strongest predictor of app adoption [11]. Conversely, respondents who rarely used other common data-collecting apps were less likely to download the COVID Alert app. This finding is consistent with a prior report in which more frequent Internet use was associated with a higher likelihood of app adoption [20]. Additionally, users who thought that the app causes unnecessary stress were significantly less likely to download it, which is consistent with prior work [5]. Demographic characteristics that were most associated with the likelihood of downloading the app included being female, in contrast to a German study [21]. Following multivariable analysis, age was not found to affect app adoption. This observation is noteworthy, as senior citizens typically make less use of digital health tools, although their adoption of technological solutions has been increasing [38,39].

Our final objective was to identify factors associated with user perceptions and knowledge of the app. Due to concerns over privacy breaches and the risk of surveillance, most digital contact tracing and exposure notification apps worldwide have opted for Bluetooth technology, which is considered much safer [26], although it may be less effective at accurately identifying high-risk contacts [40]. However, nearly half of our respondents inaccurately believed that the app uses GPS technology. Inaccurate knowledge about this key technical feature of the app significantly decreased the likelihood of downloading it. Conversely, those who self-perceived their knowledge about the app as being high were less likely to respond that it uses GPS technology and were more confident in the app’s ability to adequately protect personal data. The user’s age also has an impact on app perception, as adults aged 25 to 44 years old expressed the most concerns about risks related to user identification and inadequate data protection. These results demonstrate consistency between the respondents’ perceptions of the app and their adoption behavior. They also indicate a frequent misconception about a fundamental technical characteristic of the COVID Alert app that was specifically selected as a privacy-preserving feature.

### 4.1. Recommendations for Improved Social Aacceptability and Adoption

After combining these findings with the available literature, we have identified four areas for improving social acceptability and adoption: communication, design, access to technology, and partnership with the public.

#### 4.1.1. Communication

Our survey found significant gaps in user knowledge about the app and its functionalities. The critical gaps identified in user knowledge reveal that the costly media campaign at the launch of COVID Alert [41] did not successfully convey key information about the app. A user-informed and continued message from trusted governmental sources may have improved knowledge and, potentially, trust in the app. Indeed, the federal and provincial governments were the largest sources of influence regarding app download. In a recent study, 92% of participants demonstrated a good understanding of the COVID Alert app following an explanatory presentation of this tool [42]. Our study demonstrated the important role played by basic technical knowledge about the app in promoting trust. Media campaigns should therefore empower users by effectively sharing the information that matters most to the public. According to the Health Belief Model [37], when appropriate beliefs are held, cues to action from the government could help activate health behavior, such as promoting app downloading.

Moreover, following the principles of the privacy calculus [32], a lack of fair procedures in place to protect individual privacy (or, in this case, insufficient communication of those procedures and their benefits) could explain why half of the respondents did not download the COVID Alert app although they were inclined to use other apps.

Further communication about the app’s characteristics, functionalities, and effectiveness is also essential. More frequent and clear messaging that describes how the app works may improve trust and adoption. Messaging should be tailored to the specific concerns and potential barriers to access that affect different groups. Studying the concerns of populations that are particularly resistant to this technology may provide critical design insights and help improve communication strategies. Mobilizing health professionals and community leaders as trusted sources of information may help reach marginalized groups. Online outreach sessions, in which members of the public have an opportunity to try the app, learn about Bluetooth technology, and ask questions, may further promote trust and provide critical user feedback. Ultimately, real-time monitoring and evaluation of effectiveness are essential for building public trust in these tools [43]. While some studies have demonstrated effectiveness in particular settings, the real-life effectiveness of digital contact tracing and exposure notification remains limited two years after implementation.

#### 4.1.2. Design

Users identified app design as a focus for improvement, particularly concerning the need for additional functions that would incentivize use. Some respondents felt that the app was too passive, and recommended more opportunities for user engagement, such as the ability to follow the evolution of the pandemic locally, quickly find testing locations, and book an appointment. Some also would have preferred to receive their test result directly on the app, thus simplifying the cumbersome process of obtaining a one-time key, while still maintaining the option to share or not.

Additionally, since limited digital literacy may detract some users from adopting the app, inclusive design features are essential to make it more accessible. Translating the app into more languages than just English or French could also be an option to improve adoption, particularly in highly multicultural settings and in Indigenous communities. Indeed, a previous report has shown limited app use among people whose primary language is neither English nor French [25].

#### 4.1.3. Access to Technology

According to our results, one in seven adult Quebecers does not own a smartphone, underscoring the importance of non-smartphone-based technologies to promote access. In Australia, a hard-copy card with a unique QR code contains registered contact details and provides an alternative check-in method for customers who do not have smartphones [44]. The information contained in the QR code, along with the date and time of the visit to the premises of a business, were recorded and held for a period of 28 days by the government of New South Wales. In Singapore, authorities implemented a wearable tech version instead of making the app mandatory following a survey that highlighted citizens’ reluctance to use their mobile phones for contact tracing [45]. These initiatives could also be used in communities with limited digital literacy and device access. It is therefore important to create a balance between app effectiveness, the ability to effectively trace significant contacts, and concerns regarding data safety and privacy.

#### 4.1.4. Partnership with the Public

Respondents generally valued the importance of public involvement in the development and deployment of such apps. These findings suggest that greater upstream public engagement in the design of the COVID Alert app would have been desirable and feasible. In general, citizens should be actively involved in the development of digital health solutions [46]. Some participants highlighted the fact that public engagement in co-construction can be more difficult to achieve for tools that have a limited impact on daily life. They noted that the COVID Alert app presented few opportunities for interactions with users, and that individual incentives were harder to grasp. Clearly describing the potential public and individual impact of a digital health tool may therefore promote participation in co-construction. Academic research on these apps and related digital health innovations would also benefit from a longitudinal partnership with users, starting in the early steps of study design.

### 4.2. Study Limitations and Strengths

These findings must be interpreted in the context of some limitations. First, the results may not be generalizable to other populations. Moreover, the survey was administered during the COVID-19 pandemic, which likely affected views on related public health interventions. We also found that the self-reported proportion of app download in our study population was much higher than prior reports on the same period. Obtaining official download counts in the province of Québec has proven challenging, but the significant discrepancy noted suggests that the study population may not be representative of the full population. This may also be due to respondents’ misconceptions about the COVID Alert app, as some may have confused it with other digital tools introduced during the pandemic, such as the vaccination passport. Future research could focus on public perspectives on such apps following a description of their purpose and key characteristics, to remove any potential confusion with alternative apps. It also highlights the need for real-time, objective data on app utilization due to the discrepancy between intention to download, self-reported download, and actual download. Indeed, a public consultation held in Québec prior to the launch of the COVID Alert app also revealed a high willingness to use the app once it became available (>75% of respondents were at least likely to download it) [42]. Nonetheless, our sampling strategy and statistical corrections attempted to increase the representativeness of our study population.

Furthermore, given the relative novelty of this topic and co-design methodology, we relied on face validation following a critical synthesis of the available literature and multiple rounds of group discussions. We also performed post-hoc analyses that suggested adequate internal validity of items related to user concerns. Finally, given the cross-sectional study design, the conclusions obtained are limited to statistical associations rather than causal inference. Nonetheless, we performed multivariable logistic analyses that included literature-informed variables to best represent the potential predictors of app adoption. The directions of the most significant effects agree with the available literature. Unlike previous studies, we also identified factors associated with app knowledge and concerns, rather than limiting the analysis to predictors of app adoption. This provides additional insight into key elements for building public trust, enhancing social acceptability, and increasing the adoption of such tools.

## 5. Conclusions

This study engaged patients and members of the public in co-designing a survey on public perspectives of digital exposure notification. We identified several predictors of app download related to user characteristics and perceptions. Overall, concerns about the app were associated with lower adoption. Moreover, nearly half of the respondents inaccurately believed that the app uses GPS technology, which affected app adoption and was associated with concerns over data safety and privacy risks. Users identified governments as a main source of influence on app download, underscoring the importance of effective messaging from trusted sources. A focus on features that provide immediate value to users may further incentivize app adoption. Given the prevalence of respondents who did not own a smartphone or perceived their app literacy as being limited, digitally accessible solutions seem essential. Lastly, most respondents supported upstream public engagement in the development of digital exposure notification tools. Such engagement would also be invaluable in their deployment and evaluation.

## Figures and Tables

**Figure 1 jpm-12-00729-f001:**
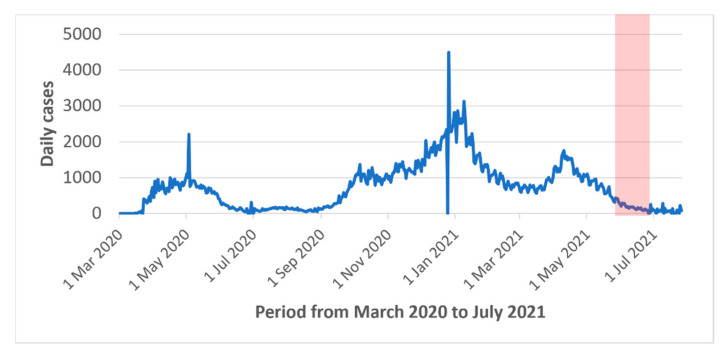
Daily new cases of COVID-19 in the province of Québec and the data collection period (in red). Data from the Government of Canada [31].

**Figure 2 jpm-12-00729-f002:**
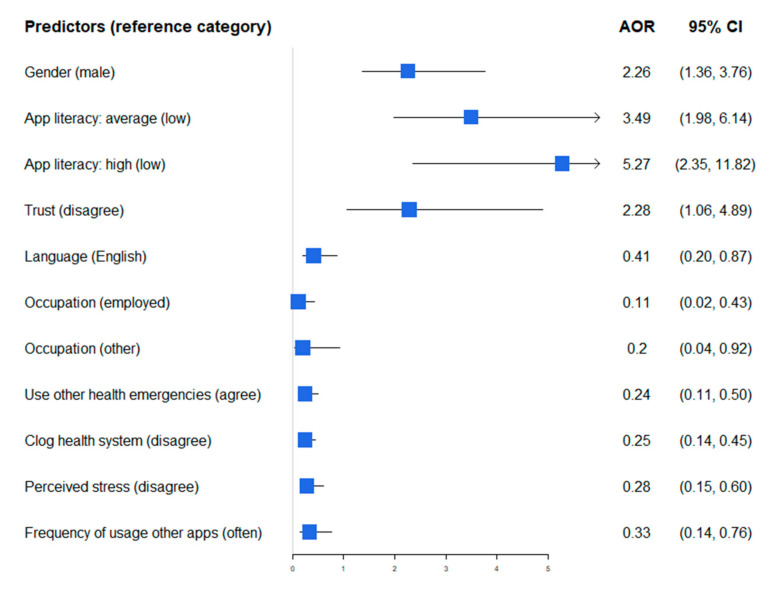
Download predictors of the COVID Alert app. Abbreviations: AOR, adjusted odds ratio; CI, confidence interval.

**Table 1 jpm-12-00729-t001:** Demographic characteristics of the survey respondents (non-weighted).

Demographic Characteristics of the COVID Alert Survey Respondents	*n* (% ^a^)
Gender	
Male	485 (50.6)
Female	471 (49.1)
Gender diverse	1 (0.1)
I prefer not to answer	2 (0.2)
Age	
18–24 years	30 (3.1)
25–34 years	106 (11.1)
35–44 years	132 (13.8)
45–54 years	171 (17.8)
55–64 years	236 (24.6)
65 years or older	284 (29.6)
Region	
Greater Montréal Area	479 (49.9)
Greater Québec Area	122 (12.7)
Other regions of Québec	353 (36.8)
Missing data	5 (0.5)
Occupation	
Working	520 (54.2)
Unemployed	39 (4.1)
Retired	335 (34.9)
Student	34 (3.5)
Disability	3 (0.3)
Other	24 (2.5)
I prefer not to answer	4 (0.4)
Education	
Without a university diploma	603 (62.9)
With a university diploma	351 (36.6)
Other	2 (0.2)
I prefer not to answer	3 (0.3)
First language	
French	823 (85.8)
English	103 (10.7)
Other	33 (3.4)
Self-perceived risk of complications from COVID-19	
No	718 (74.9)
Yes	223 (23.3)
I don’t know/I prefer not to answer	18 (1.9)
Owns a smartphone	
Yes	791 (82.5)
No	168 (17.5)
Usage frequency of common mobile apps ^b^	
Often (several times a week)	651 (82.3)
Sometimes (a few times a month)	117 (14.8)
Never	16 (2.0)
I don’t know/I prefer not to answer	7 (0.9)

^a^ Percentages before applying survey weights. ^b^ Only respondents who had a smartphone answered this question (*n* = 791).

**Table 2 jpm-12-00729-t002:** Descriptive statistics about the experience with and perspectives on the COVID Alert app and the COVID-19 pandemic.

Experience with and Perspectives on the COVID Alert App and the COVID-19 Pandemic ^a,b,c^	% (95% CI)
Have downloaded the COVID Alert app	
Yes	43.1 (39.0–47.0)
No	52.7 (48.5–57.0)
Tried to but it didn’t work	3.3 (2.2–5.0)
Don’t know/Prefer not to answer	0.9 (0.2–4.0)
COVID Alert app is…	
Activated	84.3 (79.5–88.0)
Not activated	3.3 (1.9–6.0)
Uninstalled	11.4 (8.1–16.0)
Don’t know/Prefer not to answer	1.0 (0.03–3)
Have you tested positive for COVID-19?	
Yes	3.9 (2.6–6.0)
No	95.9 (94.1–97.0)
Don’t know/Prefer not to answer	0.2 (0.05–1.0)
Did you report the positive result in the app?	
Yes	62.1 (30.0–86.0)
No	37.9 (13.5–7.0)
Don’t know/Prefer not to answer	0.0 (0)
If you had received a positive test for COVID-19, would you have reported it in the app?	
Yes	93.3 (88.0–96.0)
No	2.7 (1.4–5.0)
Don’t know/Prefer not to answer	4.0 (2.3–7.0)
Perceived risk of exposure to COVID-19	
Low	52.6 (48.8–56.0)
Moderate	37.8 (34.2–42.0)
High	8.2 (6.4–11.0)
Don’t know/Prefer not to answer	1.4 (0.6–3.0)
Perceived knowledge of the COVID Alert app	
Low	42.8 (39.0–47.0)
Average	39.2 (35.5–43.0)
High	14.1 (11.7–17.0)
Don’t know/Prefer not to answer	3.9 (2.4–6.0)
Had concerns about this type of app	
Yes	38.2 (34.5–42.0)
No	53.3 (49.0–57.0)
Don’t know/Prefer not to answer	8.51 (6.5–11.0)
The application can identify users	
Agree	28.8 (25.4–32.0)
Disagree	45.4 (41.7–49.0)
Don’t know/Prefer not to answer	25.8 (22.6–29.0)
The application locates users by GPS	
Agree	45.7 (41.9–50.0)
Disagree	31.9 (28.4–35.0)
Don’t know/Prefer not to answer	22.4 (19.5–26.0)
The app helps fight the COVID-19 pandemic	
Agree	40.0 (36.3–44.0)
Disagree	43.6 (39.8–47.0)
Don’t know/Prefer not to answer	16.4 (13.8–19.0)
The app protects personal data adequately	
Agree	36.6 (33.1–40.0)
Disagree	33.1 (29.5–37.0)
Don’t know/Prefer not to answer	30.3 (27.0–34.0)
The information collected by the app could be used to monitor the population	
Agree	46.8 (43.0–51.0)
Disagree	42.5 (38.7–46.0)
I don’t know/I prefer not to answer	10.8 (8.7–13.0)
The app only collects the information users have consented to provide	
Agree	56.6 (52.8–60.0)
Disagree	23.6 (20.5–27.0)
I don’t know/I prefer not to answer	19.7 (16.9–23.0)

^a^ Percentages after applying survey weights. ^b^ Results for the subpopulation that owns a smartphone. ^c^ Results for the subpopulation that tested positive on a COVID test. Abbreviations: CI, confidence interval; GPS, global positioning system.

## Data Availability

Not applicable.

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
