# Peer review of "Public Perspectives on Exposure Notification Apps: A Patient and Citizen Co-Designed Study"

_jpm, 2022, doi:10.3390/jpm12050729_

Round 1

Reviewer 1 Report

Any initiative involving citizens is valuable.
It has a scientific value, but also allows to design a practical tool, which increases the chances of the real use- in this case- the application.
In the era of COVID, no further theories are needed, but practical actions planned for citizens. They must be accessible and easy to use.

I would like to propose a small supplement to the content with the possibility of using such a solution (or the lack of such a possibility) in societies that are not at a high level of digitization. Even the best initiatives have their limitations and we should also talk about them; this was mentioned in the conclusions.

In conclusion: the social commitment to creating health-promoting solutions, it is a good direction and lets for the development of public health and health promotion

Author Response

Point 1: I would like to propose a small supplement to the content with the possibility of using such a solution (or the lack of such a possibility) in societies that are not at a high level of digitization. Even the best initiatives have their limitations and we should also talk about them; this was mentioned in the conclusions.

Response 1: Thank you for your suggestion. We agree, and we have added the possibility to use innovations such as a hard-copy card with a unique QR code or a wearable tech in communities with limited literacy or limited access to technology (line 462). 

Reviewer 2 Report

Please find below a review for the paper titled "Public Perspectives on Exposure Notification Apps: A Patient and Citizen Co-Designed Study" by Osmanlliu et al.

Major:
- Line 62: The percentage of voluntary app downloads (13%) seems very low based on my work and reading. We recorded 44% uptake in Australia (Garrett et al., 2021) and up to 41% in Germany (Kozyreva et al., 2021). Indeed, uptake ranged from 81% in Singapore down to 13% in India at the time of publishing the German Data. I'd recommend looking at Appendix Figure A (Kozyreva et al., 2021) for comparison across 10 countries (average uptake ~32%). I also tried to verify the 13% figure using the references provided and could not. Many of the google sheet links in [12] did not work and those I checked had no information regarding this number. Please be more specific regarding this citation as it would be very interesting and useful to readers.

- Line 80-85: This paper's case would be more strongly made using a health policy framework. Technology acceptance frameworks spring to mind, but based on the discussion, I think the authors would be best served by the factors of the health belief model (Abraham et al., 2015) - virus Severity and Susceptibility, technology Barriers (privacy concerns), and Benefits (health benefits and technology effectiveness/efficiency), Calls to action (e.g., Government messaging), and Self-efficacy. This is a well-published framework for effective health policy implementation that underpins and backs up the messages of the introduction and discussion.

- Line 245: The authors assessed the usage of other privacy encroaching apps. However, the introduction and discussion make no mention of privacy calculus (Culnan et al., 1999; i.e., how we trade off privacy for other benefits) or, more importantly in this context, the privacy paradox (Barth et al., 2017) - how people report privacy concerns but behave counter to these concerns (e.g., being privacy concerned and never updating internet passwords). It seems this is where the authors were headed, and is an interesting aspect of this literature that's worth noting in the context of public health behaviors.

- I strongly encourage the authors to make their code and data open via OSFT. There were several analysis questions that readers may have and that can be immediately addressed by looking at the analysis code. For example, how were demographic weightings completed? What modelling packages did you use? Were items adjusted appropriately for the modelling?Minor
- Line 123: Define decentralized (vs centralized) tracing for the reader. It's a privacy issue in this literature.

- Figure 1: I'd recommend a higher DPI on the figure as it's quite blurry (however, it could just be the proof I have.)

- Line 206: Rstudio isn't very important for replication, however, please report the R-software version and any packages used in the analysis (with citations to the package authors). This is much more important for replication.

- Line 208: How were sample weightings conducted? There are many methods for this. It's important here as there appears to be an oversampling of ages >55.

- Line 235: I find 38% in this context misleading. Of 959 participants, 3.9% tested positive (appx 37 people) and 38% (14 people) didn't report it via the app. There's a myriad of reasons that could explain this - they tested positive when the app was newly released or when Gov. messaging wasn't clear, or the app wasn't working. To make it clear what inference a reader should draw from this, report the actual number of people along with the percentage.

- Line 280: "Age was positively correlated..." I think this is incorrect. A positive correlation would mean older people think the app could track them, not younger (as reported).

- Line 436: A future design may assess app understanding (as done here), and then provide participants a description of the actual COVID App to gauge attitudes toward the actual app. This would remove any confusion regarding alternative apps, such as the immunity passport.

- Discussion: Consider adding a point on how Government Trust may affect app uptake. This is increasingly a factor in COVID app uptake internationally and relates to your finding that the government messaging was key to uptake. For example, we've seen this in app acceptance within the UK (Lewandowsky et al., 2021) and Taiwan (Garrett et al., 2022), but it's been found across the literature and works with this paper.Typos and grammar
Line 113 - "of innovation" -> "the innovation"References 

Abraham, C., & Sheeran, P. (2015). The health belief model. Predicting health behaviour: Research and practice with social cognition models, 2, 30-55.

Barth S, de Jong MDT. (2017). The privacy paradox—investigating discrepancies between expressed privacy concerns and actual online behavior—a systematic literature review. Telematics and Informatics. 34: 1038–1058.

Culnan, M. J., & Armstrong, P. K. (1999). Information privacy concerns, procedural fairness, and impersonal trust: An empirical investigation. Organization science, 10(1), 104-115.

Garrett, P. M., White, J. P., Lewandowsky, S., Kashima, Y., Perfors, A., Little, D. R., ... & Dennis, S. (2021). The acceptability and uptake of smartphone tracking for COVID-19 in Australia. Plos one, 16(1), e0244827.

Garrett, P. M., Wang, Y. W., White, J. P., Kashima, Y., Dennis, S., & Yang, C. T. (2022). High Acceptance of COVID-19 Tracing Technologies in Taiwan: A Nationally Representative Survey Analysis. International journal of environmental research and public health, 19(6), 3323.

Kozyreva, A., Lorenz-Spreen, P., Lewandowsky, S., Garrett, P. M., Herzog, S. M., Pachur, T., & Hertwig, R. (2021). Psychological factors shaping public responses to COVID-19 digital contact tracing technologies in Germany. Scientific reports, 11(1), 1-19.

Lewandowsky, S., Dennis, S., Perfors, A., Kashima, Y., White, J. P., Garrett, P., ... & Yesilada, M. (2021). Public acceptance of privacy-encroaching policies to address the COVID-19 pandemic in the United Kingdom. Plos one, 16(1), e0245740.

Author Response

Point 1: Line 62: The percentage of voluntary app downloads (13%) seems very low based on my work and reading. We recorded 44% uptake in Australia (Garrett et al., 2021) and up to 41% in Germany (Kozyreva et al., 2021). Indeed, uptake ranged from 81% in Singapore down to 13% in India at the time of publishing the German Data. I'd recommend looking at Appendix Figure A (Kozyreva et al., 2021) for comparison across 10 countries (average uptake ~32%). I also tried to verify the 13% figure using the references provided and could not. Many of the google sheet links in [12] did not work and those I checked had no information regarding this number. Please be more specific regarding this citation as it would be very interesting and useful to readers.

Response 1: Thank you for your comment. We have consulted the references you have provided, and we have included the data in Appendix Figure A of Kozyreva et al. 2021 in the manuscript instead of those in the MIT COVID Tracing Tracker (line 61). We agree that this gives a more realistic portrait of the use exposure notifications apps.

Point 2: Line 80-85: This paper's case would be more strongly made using a health policy framework. Technology acceptance frameworks spring to mind, but based on the discussion, I think the authors would be best served by the factors of the health belief model (Abraham et al., 2015) - virus Severity and Susceptibility, technology Barriers (privacy concerns), and Benefits (health benefits and technology effectiveness/efficiency), Calls to action (e.g., Government messaging), and Self-efficacy. This is a well-published framework for effective health policy implementation that underpins and backs up the messages of the introduction and discussion.

Response 2: Thank you for your suggestion. We have added the health belief model in our discussion to explain how cues to action from the governments could help activate behavior to download the app (lines 335 and 390).

Point 3: Line 245: The authors assessed the usage of other privacy encroaching apps. However, the introduction and discussion make no mention of privacy calculus (Culnan et al., 1999; i.e., how we trade off privacy for other benefits) or, more importantly in this context, the privacy paradox (Barth et al., 2017) - how people report privacy concerns but behave counter to these concerns (e.g., being privacy concerned and never updating internet passwords). It seems this is where the authors were headed, and is an interesting aspect of this literature that's worth noting in the context of public health behaviors

Response 3: Thank you for the suggestion. We have incorporated both the privacy calculus and the privacy paradox in the discussion (lines 315 and 393). We agree that these principles enriched the interpretations of our results as it offers possible explanations as to why even respondents that did not download the COVID Alert app are using other apps that collects data (contrary to the COVID Alert App).

Point 4: I strongly encourage the authors to make their code and data open via OSFT. There were several analysis questions that readers may have and that can be immediately addressed by looking at the analysis code. For example, how were demographic weightings completed? What modelling packages did you use? Were items adjusted appropriately for the modelling? Minor

Response 4: Thank you for your suggestion. We agree and have made the code public via OSF. We have added the information of modelling packages as well as the link to the code in supplementary materials (document S2).

Point 5: Line 123: Define decentralized (vs centralized) tracing for the reader. It's a privacy issue in this literature.

Response 5: Thank you for the suggestion. We have added the distinction between decentralized and centralized apps (line 127).

Point 6: Figure 1: I'd recommend a higher DPI on the figure as it's quite blurry (however, it could just be the proof I have.)

Response 6: Thank you for your recommendation. We have increased the DPI of Figure 1.

Point 7: Line 206: Rstudio isn't very important for replication, however, please report the R-software version and any packages used in the analysis (with citations to the package authors). This is much more important for replication.

Response 7: Thank you for your suggestion. The R-software version is indicated in line 213. We have added the information in supplementary materials (document S2).

Point 8: Line 208: How were sample weightings conducted? There are many methods for this. It's important here as there appears to be an oversampling of ages >55.

Response 8: Thank you for your comment. Sample weightings were processed by SOM, an independent research firm. The variables used to calculate sample weights are included at line 203. Moreover, document S3 includes the detailed methodology provided by SOM. Percentages in Table 1 correspond to non-weighted data and may explain the oversampling you have noticed. The sample weighting process addressed this issue.

Point 9: Line 235: I find 38% in this context misleading. Of 959 participants, 3.9% tested positive (appx 37 people) and 38% (14 people) didn't report it via the app. There's a myriad of reasons that could explain this - they tested positive when the app was newly released or when Gov. messaging wasn't clear, or the app wasn't working. To make it clear what inference a reader should draw from this, report the actual number of people along with the percentage.

Response 9: Thank you for your suggestion. We agree that the sentence can be misleading. We have added the number (n=4), but it has to be considered that the results presented in this section are based on weighted data.

Point 10: Line 280: "Age was positively correlated..." I think this is incorrect. A positive correlation would mean older people think the app could track them, not younger (as reported).

Response 10: Thank you for your comment. The age variable is categorized by the following: 18-24 years, 25-34 years, 35-44 years, 45-54 years, 55-64 years, 65 years and over. In this estimation, the base category is 18-24 years old. To be more precise, the comparison should be made specifically with this age group. We have made the necessary adjustment in line 285.

Point 11: Line 436: A future design may assess app understanding (as done here), and then provide participants a description of the actual COVID App to gauge attitudes toward the actual app. This would remove any confusion regarding alternative apps, such as the immunity passport.

Response 11: Thank you for your suggestion. We have added specific suggestions for future research that include providing a description of the app’s purpose and key characteristics to remove any potential confusion with alternative apps (line 462).

Point 12: Discussion: Consider adding a point on how Government Trust may affect app uptake. This is increasingly a factor in COVID app uptake internationally and relates to your finding that the government messaging was key to uptake. For example, we've seen this in app acceptance within the UK (Lewandowsky et al., 2021) and Taiwan (Garrett et al., 2022), but it's been found across the literature and works with this paper.

Response 12: Thank you for this suggestion, we agree and have added a point on how government trust may have affected app uptake in the discussion (line 332).

Point 13: Typos and grammar Line 113 - "of innovation" -> "the innovation"

Response 13: Thank you for noticing this mistake, we have made the necessary correction in the revised manuscript (line 117).